

# Automatic sleep staging by a hybrid model based on deep 1D-ResNet-SE and LSTM with single-channel raw EEG signals

Weiming Li and Junhui Gao

Shanghai Nuanhe Brain Technology Co. Ltd., Shanghai, China

## ABSTRACT

Sleep staging is crucial for assessing sleep quality and diagnosing sleep disorders. Recent advances in deep learning methods with electroencephalogram (EEG) signals have shown remarkable success in automatic sleep staging. However, the use of deeper neural networks may lead to the issues of gradient disappearance and explosion, while the non-stationary nature and low signal-to-noise ratio of EEG signals can negatively impact feature representation. To overcome these challenges, we proposed a novel lightweight sequence-to-sequence deep learning model, 1D-ResNet-SE-LSTM, to classify sleep stages into five classes using single-channel raw EEG signals. Our proposed model consists of two main components: a one-dimensional residual convolutional neural network with a squeeze-and-excitation module to extract and reweight features from EEG signals, and a long short-term memory network to capture the transition rules among sleep stages. In addition, we applied the weighted cross-entropy loss function to alleviate the class imbalance problem. We evaluated the performance of our model on two publicly available datasets; Sleep-EDF Expanded consists of 153 overnight PSG recordings collected from 78 healthy subjects and ISRUC-Sleep includes 100 PSG recordings collected from 100 subjects diagnosed with various sleep disorders, and obtained an overall accuracy rate of 86.39% and 81.97%, respectively, along with corresponding macro average F1-scores of 81.95% and 79.94%. Our model outperforms existing sleep staging models in terms of overall performance metrics and per-class F1-scores for several sleep stages, particularly for the N1 stage, where it achieves F1-scores of 59.00% and 55.53%. The kappa coefficient is 0.812 and 0.766 for the Sleep-EDF Expanded and ISRUC-Sleep datasets, respectively, indicating strong agreement with certified sleep experts. We also investigated the effect of different weight coefficient combinations and sequence lengths of EEG epochs used as input to the model on its performance. Furthermore, the ablation study was conducted to evaluate the contribution of each component to the model's performance. The results demonstrate the effectiveness and robustness of the proposed model in classifying sleep stages, and highlights its potential to reduce human clinicians' workload, making sleep assessment and diagnosis more effective. However, the proposed model is subject to several limitations. Firstly, the model is a sequence-to-sequence network, which requires input sequences of EEG epochs. Secondly, the weight coefficients in the loss function could be further optimized to balance the classification performance of each sleep stage. Finally, apart from the channel attention mechanism, incorporating more advanced attention mechanisms could enhance the model's effectiveness.

Corresponding author
Weiming Li, lwm921122@gmail.com

## INTRODUCTION

Sleep plays a crucial role in maintaining optimal cognitive and emotional functioning in daily life. Unfortunately, sleep-related disorders can negatively impact sleep quality, and pose a significant burden on public health worldwide. Therefore, it is essential to accurately estimate sleep quality to ensure prompt medical interventions when necessary. Sleep staging is a critical aspect of evaluating sleep quality and diagnosing sleep disorders, such as insomnia and obstructive sleep apnea (*Wulff, Gatti & Wettstein JG, 2010*). Polysomnography (PSG) is widely regarded as the gold standard for sleep staging, involving the recording of several electrophysiological signals, such as electroencephalogram (EEG), electrooculogram (EOG), electromyogram (EMG), and electrocardiogram (ECG) (*Keenan, 2005*). During manual sleep staging, certified sleep technicians visually examine PSG signals and label sleep stages in successive 30-second epochs, following either the Rechtschaffen and Kales (R&K) Rules or the American Academy of Sleep Medicine (AASM) Manual (*Moser et al., 2009*). The R&K Rules divide sleep into wake (W), rapid eye movement (REM), and non-REM (NREM) stages, where the NREM stage is further subdivided into S1, S2, S3, and S4, based on the depth of sleep (*Wolpert, 1969*). The AASM standard subsequently modifies the R&K Rules by merging S3 and S4 into a single stage, N3, and subdividing N1, N2, and N3 as subdivisions of the NREM. Consequently, the AASM standard classifies sleep stages into W, N1, N2, N3, and REM (*Berry et al., 2017*).

However, manual sleep staging is a laborious, time-consuming, and subjective process with significant inter- and intra-rater variability (*Danker-hopfe et al., 2009*). It typically takes up to two hours for a sleep expert to score an overnight PSG recording of eight hours, with an inter-rater agreement of approximately 83% (*Rosenberg & Van Hout, 2013*). Thus, developing reliable and accurate automatic sleep-staging algorithms is highly promising. Furthermore, PSG, which involves recording multiple electrophysiological signals, is expensive, complicated, and impractical for longitudinal sleep monitoring in home settings. PSG can also cause physical discomfort and mental stress, leading to poor sleep quality and disruption of regular sleep patterns (*Wang et al., 2021*). Consequently, there is a growing interest in developing portable, wireless, and wearable devices equipped with low-cost sensors attached to a single location, such as the Fpz channel in the frontal region of the brain, to acquire single-channel EEG signals, facilitating comfortable and convenient sleep monitoring (*Gao et al., 2022*).

Numerous studies have employed a variety of machine learning algorithms with EEG signals for sleep staging (*Vallat & Walker, 2021*; *Hussain et al., 2022*) and other applications, such as disease detection (*Islam et al., 2022*). Nevertheless, these approaches involve feature engineering to extract relevant features in time, frequency, time-frequency, or nonlinear domains, which requires domain knowledge and may result in information loss.

In recent years, advances in deep learning methods have shown remarkable success in many applications using EEG signals. Various automatic sleep-staging models have

been developed based on end-to-end deep learning methods that use single-channel raw EEG signals. These models have demonstrated comparable performance to human sleep experts in terms of scoring accuracy. Many of these models utilize convolutional neural networks (CNN) (*Krizhevsky, Sutskever & Hinton, 2017*) to extract relevant features from raw EEG signals, and various recurrent neural network (RNN) variants to learn temporal dependencies and transition rules among sleep stages. For instance, *Mousavi, Afghah & Acharya (2019)* proposed SleepEEGNet, a sequence-to-sequence model that uses a CNN architecture with two sections to extract temporal and frequency information from EEG signals, followed by an encoder–decoder architecture incorporating a bidirectional recurrent neural network (Bi-RNN) and an attention mechanism to capture and emphasize the temporal information. DeepSleepNet, introduced by *Supratak et al., (2017)*, utilizes two CNNs with different convolutional filter sizes for representation learning, followed by two bidirectional long short-term memory (Bi-LSTM) layers for sequence residual learning. TinySleepNet (*Supratak & Guo, 2020*), another model proposed by Supratak and his colleagues, uses fewer convolutional layers and one-directional LSTM layers to reduce computational resources. IITNet, proposed by *Seo et al. (2020)*, employs CNNs with residual connections to extract distinctive features from EEG signals in an intra-epoch manner, and adopts Bi-LSTM to model inter-epoch contextual information. AttnSleep, developed by *Eldele et al. (2021)*, is an attention-based deep learning architecture that leverages multi-resolution CNNs to extract low- and high-frequency features of EEG signals, while emplying an adaptive feature recalibration technique to improve the quality of the extracted features by modeling the inter-dependencies between them, and uses a multi-head attention mechanism to capture the temporal relationships among sleep stages. In another study, *Liu et al. (2022)* employed parallel residual neural networks with improved channel and spatial feature attention units for multi-scale feature extraction of EEG sleep signals, while a bi-directional gated recurrent unit (Bi-GRU) was utilized to determine the dependence between sleep stages.

Among the above-mentioned deep learning models, multi-layer architectures of one-dimensional convolutional neural networks (1D-CNNs) have been shown to be effective in extracting informative features from EEG signals for sleep staging. The depth of the CNNs has a significant positive influence on feature expression ability (*Simonyan & Zisserman, 2014*). However, deep neural networks are prone to gradient disappearance and explosion problems (*He & Sun, 2015*). To address this issue, *He et al. (2016)* introduced a residual convolutional neural network (ResNet), which incorporates an identical connection to alleviate the gradient degradation problem in image recognition tasks effectively. Therefore, 1D-ResNet is a neural network model specially designed for processing one-dimensional sequential data, such as EEG signals. Meanwhile, it leverages residual connections to facilitate the flow of information and alleviate the vanishing gradient problem. Additionally, EEG signals can be affected by various background noises, and the non-stationary nature and low signal-to-noise ratio of EEG signals can degrade feature representation. To overcome these challenges, attention mechanisms can be used for feature recalibration (*Vaswani et al., 2017*). In particular, the Squeeze-and-Excitation (SE) module is a channel-wise attention mechanism that enhances the representational

power of neural networks by explicitly modeling interdependencies between channels. It adaptively recalibrates channel-wise feature responses by learning channel-specific importance weights. This allows the model to dynamically focus on more informative channels and suppress less relevant ones. Furthermore, the SE module can be integrated into CNN architectures to improve model performance without changing the shape of existing models (*Hu, Shen & Sun, 2018*). When the 1D-ResNet is combined with the SE module, it leverages the strengths of both architectures. The residual connections in the 1D-ResNet facilitate the propagation of gradients, enabling efficient training and reducing the risk of overfitting. Meanwhile, the SE module helps the network to capture and emphasize important channel-level features, enhancing discriminative capabilities. Taking together, the integration of the 1D-ResNet block with the SE module demonstrates promising potential in EEG-based sleep stage classification tasks.

The number of samples for each sleep stage in sleep staging tasks is typically unbalanced due to variations in the time spent in different sleep stages by individuals (*Phan & Mikkelsen, 2022*). In addition, the difficulty level of each sleep stage being distinguished varies. Various approaches have been proposed to address the class imbalance issue in sleep staging. One common technique is class-balanced random sampling (*Tsinalis et al., 2016*), where the dataset is sampled in a manner that ensures each sleep stage is represented proportionally. Another approach, class-balance training set design (*Supratak et al., 2017*), involves designing a balanced training set by carefully selecting and augmenting data from each sleep stage. Synthetic minority oversampling technique (SMOTE) (*Chawla et al., 2002*) has also been utilized to generate synthetic samples for minority classes, thus re-balancing the dataset. Data augmentation techniques (*Fawaz et al., 2018*), such as introducing variations or transformations to existing samples, have been employed as well. However, applying these methods to sleep staging tasks may introduce challenges in preserving the actual order of sleep stages and capturing the temporal dependency features and sleep-stage transition rules. A potential alternative is to utilize a loss function specifically designed for imbalanced data, such as the weighted cross-entropy loss function (*Lin et al., 2017*). This loss function assigns different weights to each class, providing higher emphasis on the minority classes and effectively addressing the class imbalance issue.

In light of the challenges mentioned earlier and inspired by previous studies, in this study, we proposed a new lightweight sequence-to-sequence deep learning model, called 1D-ResNet-SE-LSTM, for classifying sleep stages into five categories using single-channel raw EEG signals. Our proposed model consists of two major components. The first is the 1D-ResNet-SE block, which extracts and reweights features from EEG signals while addressing the gradient degradation problem caused by network deepening through its residual network architecture. The second is the LSTM network, which captures the temporal dependencies and transition rules among sleep stages. In addition, the weighted cross-entropy loss function was applied to mitigate the class imbalance problem. The performance of our model was evaluated on two publicly available datasets, Sleep-EDF Expanded and ISRUC-Sleep. Overall, our model aims to improve current deep-learning-based sleep staging models from four perspectives:

1. A sequence-to-sequence model framework is employed, consisting of three 1D-ResNet blocks and one LSTM network. The 1D-ResNet block extracts multi-level features from single-channel raw EEG signals, while the LSTM network captures the transition rules among sleep stages;

2. The SE module is incorporated into the 1D-ResNet block to recalibrate the extracted features and emphasize critical information for discriminating between different sleep stages;

3. The weighted cross-entropy loss function is utilized to address the class imbalance problem, with weight parameters determined based on the number of samples in each class and the difficulty level of each class being distinguished, leading to a significant improvement in the performance of the N1 stage;

4. Our model is designed to be as lightweight as possible, thereby saving computational resources and making it more practical for real-world applications.

## MATERIALS AND METHODS

### Data

Portions of this text were previously published as part of a preprint (*Li & Gao, 2023*).

To assess the proposed model's effectiveness and generalizability, we conducted evaluations on two publicly available datasets, namely Sleep-EDF Expanded (*Goldberger et al., 2000*) and ISRUC-Sleep (*Khalighi et al., 2016*). These datasets differ in various aspects, such as recording settings, subject health status, EEG channels, sampling rate, data preprocessing, and annotation criteria, thereby providing an opportunity to test the robustness and adaptability of the model.

The Sleep-EDF Expanded dataset comprises 197 whole-night PSG recordings derived from two studies: one examining the impact of age on sleep in healthy subjects (SC: Sleep Cassette), and the other investigating the effect of temazepam on sleep (ST: Sleep Telemetry). For this study, we selected the SC dataset, which consists of 153 overnight PSG recordings collected from 78 healthy subjects (37 males and 41 females) between the ages of 25 and 101 years. Each PSG recording includes two EEG signals acquired from Fpz-Cz and Pz-Oz channels, one horizontal EOG, one chin EMG, and one oro-nasal respiration. The EEG signals are sampled at a rate of 100 Hz. The PSG recordings were segmented into non-overlapping 30-second epochs and manually annotated by a certified sleep technician following the R&K Rules, classifying each epoch into one of eight stages: W, S1, S2, S3, S4, REM, M (Movement), and UNKNOWN stages.

The ISRUC-Sleep dataset comprises overnight PSG recordings from 118 subjects diagnosed with various health conditions, including healthy, sick, and under treatment. Subgroup I, which includes 100 PSG recordings collected from 100 subjects (55 males and 45 females) diagnosed with various sleep disorders aged between 20 and 85 years, was used in this study. Each PSG recording contains multiple signals from six EEG channels (C3_A2, C4_A1, F3_A2, F4_A1, O1_A2, and O2_A1), two EOG channels (LOC_A2 and ROC_A1), and three EMG channels (X1, X2, and X3), with the sampling rate of 200 Hz. Two sleep experts independently annotated each PSG recording based on the AASM standard, with

**Table 1  Sample sizes and corresponding percentages of each sleep stage for two datasets used in this study.**

| Dataset | W | N1 | N2 | N3 | REM | Total |
|---|---|---|---|---|---|---|
| Sleep-EDF Expanded | 69,824 | 21,522 | 69,132 | 13,039 | 25,835 | 199,352 |
| | 35.0% | 10.8% | 34.7% | 6.5% | 13.0% | 100.0% |
| ISRUC-Sleep | 20,098 | 11,062 | 27,511 | 17,251 | 11,265 | 87,187 |
| | 23.0% | 12.7% | 31.6% | 19.8% | 12.9% | 100.0% |

each 30-second epoch categorized into one of five different sleep stages: W, N1, N2, N3, and REM.

## Data preprocess

Regarding the Sleep-EDF Expanded dataset, the M and UNKNOWN stages were excluded from analysis due to their limited sample sizes and lack of belonging to any specific sleep stage. Following the AASM Manual, the S3 and S4 stages were merged into a single stage, S3, resulting in a five-class classification problem for sleep staging, to facilitate a comparison of the scoring performance with previous studies. To emphasize the sleep stages, only 30 min of wake periods before and after the sleep periods were selected for analysis. The Fpz-Cz EEG signals were selected for model evaluation due to their high performance and practicality in home environments (*Supratak et al., 2017*; *Imtiaz, 2021*). Notably, no denoising or filtering preprocessing was applied to the raw EEG signals in line with previous studies (*Supratak et al., 2017*; *Supratak & Guo, 2020*).

On the other hand, for the ISRUC-Sleep dataset, preprocessing steps have already been performed on the EEG signals to eliminate unwanted noise and DC offset, thereby improving the quality and signal-to-noise ratio of the data. A notch filter was used to remove the 50 Hz powerline noise, and the signals were bandpass-filtered between 0.3 Hz and 35 Hz. The EEG channel F3_A2 was chosen for model evaluation, and the EEG signals were resampled at 100 Hz.

Table 1 summarizes the sample sizes and corresponding percentages of each sleep stage for both datasets used in this study. The Sleep-EDF Expanded dataset comprises 199,352 samples, with W, N1, N2, N3, and REM stages accounting for 35.0%, 10.8%, 34.7%, 6.5%, and 13.0% of the samples, respectively. The ISRUC-Sleep dataset includes 87,187 samples, with W, N1, N2, N3, and REM stages representing 23.0%, 12.7%, 31.6%, 19.8%, and 12.9% of the samples, respectively.

## The proposed model 1D-ResNet-SE-LSTM
### The SE module

The architecture of the SE module is depicted in Fig. 1. The SE module is designed to calibrate channel-wise features by considering inter-channel dependencies, consisting of three operations, namely squeezing, excitation, and scaling. Specifically, multiple feature maps produced by the convolutional operation serve as input to the SE module. In the squeezing operation, the feature maps are squeezed by the global average pooling (GAP) layer (*Szegedy et al., 2015*) to create a weight vector. Next, in the excitation operation, the weight vector is passed through two successive fully connected (FC) layers, each of which is

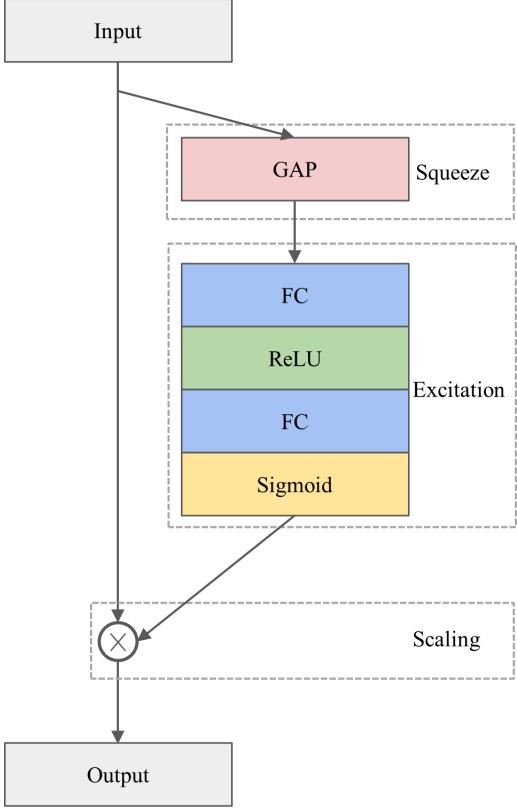

**Figure 1  The architecture of the SE module.**

activated by a rectified linear unit (ReLU) (*Nair & Hinton, 2010*) and a sigmoid function, respectively. These procedures introduce a nonlinear transformation and provide a smooth gating calculation, resulting in a channel weight vector. Finally, the channel weight vector is multiplied by the input feature maps to produce the final output in the scaling operation. In other words, the input feature maps are reweighted by the channel weight vector.

### The 1D-ResNet-SE block

The proposed 1D-ResNet-SE block, as illustrated in Fig. 2, consists of two successive one-dimensional convolutional (Conv1D) layers. The first Conv1D layer is followed by a batch normalization (BN) layer and a ReLU activation function layer, while the second Conv1D layer is followed by a BN layer, an SE module, and a ReLU activation function layer sequentially. The Conv1D layer is utilized to extract relevant features from the raw data (*Kiranyaz et al., 2021*), while the BN layer is employed to standardize the data batch and improve the generalization capability of the model, also addressing the common issue of vanishing gradient in deep learning (*Ioffe & Szegedy, 2015*). The ReLU activation function is a nonlinear function that enhances the model's expression ability. The SE module is incorporated after the second BN layer to calibrate the features.

To tackle the challenges of gradient dispersion and explosion, a shortcut connection is established between the input $x$ and output of the SE module. The shortcut path involves

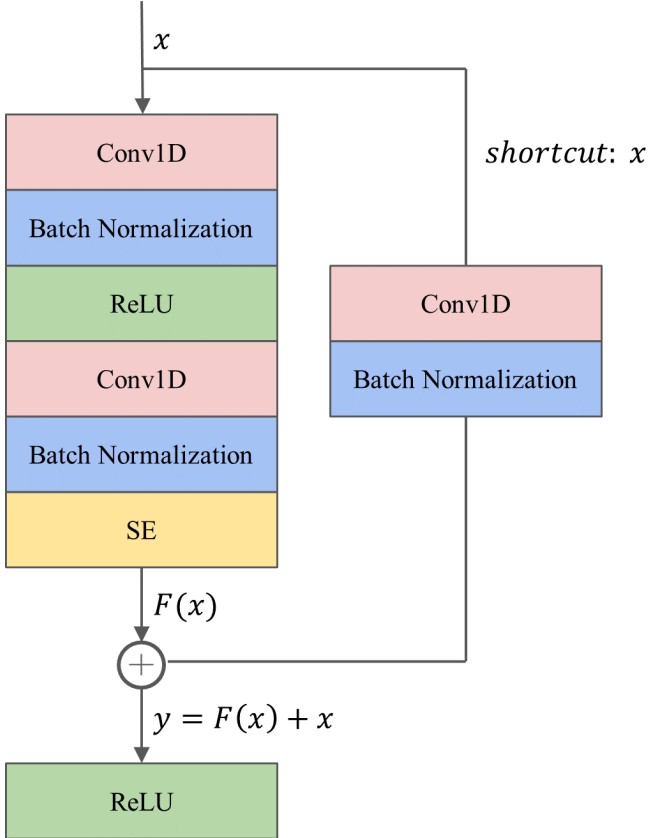

**Figure 2** **The architecture of the 1D-ResNet-SE block.**

a Conv1D layer with a convolutional kernel size of one to match the input and output dimensions, followed by a BN layer. For a network with input $x$, the underlying mapping is denoted as $H(x)$, and the residual mapping is $F(x) = H(x) - x$. The rationale behind the residual network is to enable the neural network to fit the residual mapping instead of learning the underlying mapping. The output y from the shortcut operation is computed as $y = F(x) + x$.

### LSTM network

The LSTM network, a variant of RNNs, was proposed by *Hochreiter & Schmidhuber (1997)*. This network is specifically designed to capture long-term dependencies and address the issues of vanishing and exploding gradients commonly encountered during long-sequence learning. In contrast to ordinary RNNs, the LSTM network includes the memory cell instead of the recurrent node.

At each time step, the LSTM network inputs the current time step input $X_t$ and the hidden state of the previous time step $H_{t-1}$. The network comprises three gates, namely forget, input, and output gates, responsible for information selection. These gates produce output values, $F_t$, $I_t$, and $O_t$, respectively, using a fully connected layer with the sigmoid $\sigma$ activation function, ranging between 0 and 1. Among these gates, the forget gate decides

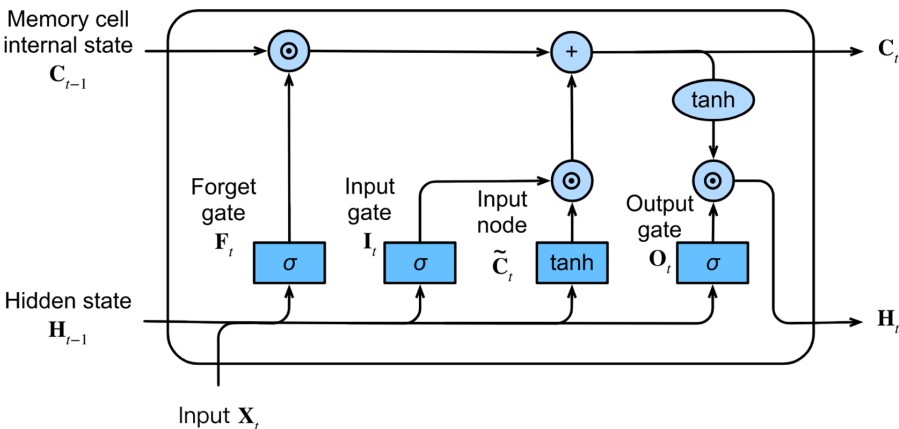

**Figure 3** The architecture of the LSTM network.

which information from the previous memory cell's internal state $C_{t-1}$ needs to be removed or retained. The input gate determines which information in the input node $\widetilde{C}_t$ should be added to the current memory cell's internal state $C_t$, where the input node is obtained from a fully connected layer with the tanh activation function that ranges between $-1$ and $1$. The output gate decides which information in the updated cell state should be included in the output hidden state at the current time step $H_t$. Figure 3 illustrates the LSTM network and its internal architecture.

### The proposed model

The proposed model, 1D-ResNet-SE-LSTM, aims to predict a sequence of sleep stages in a many-to-many scheme from a series of EEG epochs. The input to the model consists of an ordered set of $N$ single-channel EEG epochs, denoted by $\{x_1, \ldots, x_N\}$, where $x_i \in \mathbb{R}^{E_s \times F_s}$, $x_i$ represents the $i$-th EEG epoch, and $1 \leq i \leq N$, with $E_s$ representing the duration of the EEG epoch in seconds, and $F_s$ denoting the EEG signals' sampling rate. The model generates an equally long sequence of predicted sleep stages $\{\hat{y}_1, \ldots, \hat{y}_N\}$, where $\hat{y}_i$ corresponds to the predicted sleep stage for $x_i$, and $\hat{y}_i \in \{0, 1, 2, 3, 4\}$ denotes the five sleep stages W, N1, N2, N3, and REM, respectively. The value of $N$ is set to 15, $E_s$ is 30, and $F_s$ is 100 for this study.

The model's architecture consists of three sequential 1D-ResNet-SE blocks, each with 32, 64, and 128 convolutional kernels, and convolutional kernel sizes of 3, 5, and 7, respectively. The strides are set to 1, and no padding is applied. The first two blocks are followed by a max-pooling layer that reduces the feature map size by a factor of four and preserves essential features, while the last block is followed by a GAP layer that averages each feature map. A dropout layer with a regularization probability of 50% is applied to improve the model's generalization ability (*Srivastava et al., 2014*). The 1D-ResNet-SE blocks serve as the epoch-wise feature extractor to convert the EEG epoch $x_i$ into the feature vector $a_i$ using learnable parameters $\theta_r$, as given by Eq. (1).

$$a_i = ResNet_{\theta_r}(x_i) \tag{1}$$

As a result, an input sequence of EEG epochs is transformed into a sequence of feature vectors, which is then fed into the LSTM layer as a series of inputs, where each feature vector corresponds to a specific time step in the sequence. The LSTM layer has 64 hidden units and is responsible for learning transition rules among sleep stages. Given $N$ feature vectors $\{a_1, \ldots, a_N\}$ arranged sequentially, the LSTM layer processes the $i$-th feature vector $a_i$ using Eq. (2):

$$h_i, c_i = LSTM_{\theta_s}(h_{i-1}, c_{i-1}, a_i) \tag{2}$$

where $LSTM_{\theta_s}$ represents the LSTM layer processing the sequence of feature vectors $a_i$, $\theta_s$ is the learnable parameters of the LSTM, $h_i$ and $c_i$ are vectors of hidden and cell states of the LSTM layer after processing the feature vector $a_i$, $h_{i-1}$, and $c_{i-1}$ are the hidden and cell states from the previous time step, $h_0$ and $c_0$ are the initial hidden and cell states that are set to 0.

Lastly, a Softmax layer is employed for the five-class classification, producing a five-dimensional output vector corresponding to the probabilities of each class (W, N1, N2, N3, and REM) that the given EEG epoch belongs to. The sleep stage label for each epoch is determined by the maximum class probability using the argmax function, resulting in the model generating a predictive sleep stage sequence. The model's schematic representation is shown in Fig. 4.

## Training
### Loss function
To address the issue of imbalanced class distribution in the dataset, the weighted cross-entropy loss function was utilized in this study. The weight coefficients are determined based on the number of samples in each class and the difficulty level of distinguishing each class. The N1 and N3 stages, which have fewer samples than the other stages, are considered minority classes. Moreover, the N1 stage is particularly challenging to classify due to its transitional nature from wakefulness to sleeping (*Phan et al., 2018*), whereas the N3 stage has distinctive characteristics, such as significantly higher amplitude, making it easier to identify. To account for these imbalances, a weight coefficient of 1.5 was assigned to the N1 stage, while the weight coefficients for the remaining four stages were set to 1. The weighted cross-entropy loss function is formulated as Eq. (3).

$$Loss = -\sum_{i}^{C} w_i * y_i * \log(\hat{y}_i) \tag{3}$$

where $y_i$ and $\hat{y}_i$ represent the actual and predicted labels, respectively, for each class $i$ in $C$ classes, and $w_i$ denotes the weight coefficient assigned to each class.

### Training setup
The dataset was partitioned into training and test sets with proportions of 90% and 10%, respectively, while 10% of the training set was extracted as the validation set. To minimize the loss function, the adaptive moment estimation (Adam) optimizer (*Kingma & Ba, 2014*) was employed, with an initial learning rate of 0.001. The step size was reduced by a factor

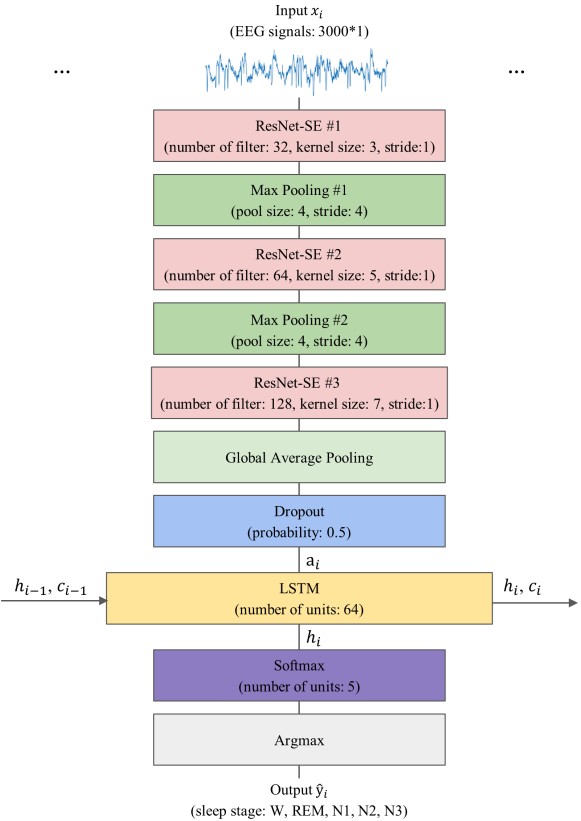

**Figure 4** **The overall architecture of the proposed model.**

of 10 after every 10 epochs if the validation loss did not decrease. To prevent overfitting, early stopping was implemented when the validation loss remained unchanged for 20 epochs (*Basheer & Hajmeer, 2000*). The model with the highest validation accuracy was selected as the final model based on the model checkpoints. A training batch size of 16 was used, and the model was trained for 100 epochs. Finally, the performance of the model was assessed using the test set.

### Model implement environment

We implemented our model using the TensorFlow and Keras frameworks in a Python 3 environment running on an Ubuntu 20.04 system. The computation was performed on an NVIDIA GRID T4-4Q GPU with 4GB of GPU memory, hosted on the Alibaba Cloud Elastic Compute Service.

## Performance evaluation
### Per class evaluation

The model' performance was evaluated by calculating precision, recall, and F1-score for each class on the test set. These metrics are defined as follows:

$$Precision = \frac{TP}{TP + FP}, \tag{4}$$

$$Recall = \frac{TP}{TP + FN}, \tag{5}$$

$$F1 - score = \frac{2 \times TP}{2 \times TP + FP + FN}, \tag{6}$$

where TP, FP, TN, and FN refer to the number of true positive, false positive, true negative, and false negative predictions, respectively.

### *Overall evaluation*

The model's overall performance was evaluated using the overall accuracy (Acc), as defined in Eq. (7), where $C = 5$ represents the number of target classes, $TP_i$ denotes the number of true positives in class $i$, and $M$ is the total number of EEG epochs.

$$Acc = \frac{\sum_{i=1}^{C} TP_i}{M} \tag{7}$$

To account for the imbalanced nature of the datasets, the macro average F1-score (MF1) was also used, which is defined in Eq. (8), where $F1_i$ is per-class F1-score for class $i$.

$$MF1 = \frac{\sum_{i=1}^{C} F1_i}{C} \tag{8}$$

Moreover, Cohen's kappa coefficient ($\kappa$) (*Cohen, 1960*) was computed to assess the degree of agreement between the model's classification results and those of human sleep experts, as given by Eq. (9):

$$\kappa = \frac{p_o - p_e}{1 - p_e} \tag{9}$$

where $p_o$ denotes the number of samples the raters agree on and is divided by the total number of samples, $p_e$ represents the probability of chance agreement.

In addition, the confusion matrix and its normalized version were generated to provide insight into the classification performance.

## RESULTS

### Classification performance

Table 2 presents the results of the model's classification performance on the Sleep-EDF Expanded dataset. The table lists the per-class precision, recall, and F1-score for each sleep stage, and the overall Acc, MF1, and $\kappa$ coefficient. The model exhibits the highest precision, recall, and F1-score of 94.90%, 94.33%, and 94.62%, respectively, for the W stage. In contrast, the lowest precision, recall, and F1-score of 59.08%, 58.92%, and 59.00%, respectively, are observed for the N1 stage. These findings suggest that the model better identifies the W stage and struggles with recognizing the N1 stage. The model exhibits intermediate performance for N2, N3, and REM stages, with F1-scores of 87.95%, 82.79%, and 85.38%, respectively. The overall Acc and MF1 are 86.39% and 81.95%, respectively, indicating the model's promising performance in sleep staging. Furthermore, the $\kappa$ value of

**Table 2  Classification performance on the Sleep-EDF Expanded dataset.**

| Class | Per-class performance | | | Overall performance | | |
|---|---|---|---|---|---|---|
| | Precision | Recall | F1-score | Acc | MF1 | $\kappa$ |
| W | 94.90 | 94.33 | 94.62 | | | |
| N1 | 59.08 | 58.92 | 59.00 | | | |
| N2 | 86.54 | 89.42 | 87.95 | 86.39 | 81.95 | 0.812 |
| N3 | 87.78 | 78.34 | 82.79 | | | |
| REM | 86.20 | 84.58 | 85.38 | | | |

**Notes.**
[1]The numerical values in this table and subsequent ones are expressed in percentage, with the exception of the $\kappa$ index.

**Table 3  Classification performance on the ISRUC-Sleep dataset.**

| Class | Per-class performance | | | Overall performance | | |
|---|---|---|---|---|---|---|
| | Precision | Recall | F1-score | Acc | MF1 | $\kappa$ |
| W | 93.06 | 85.25 | 88.98 | | | |
| N1 | 55.30 | 55.76 | 55.53 | | | |
| N2 | 80.92 | 84.15 | 82.50 | 81.97 | 79.94 | 0.766 |
| N3 | 90.59 | 87.94 | 89.25 | | | |
| REM | 80.49 | 86.65 | 83.46 | | | |

0.812, which falls within the almost perfect agreement range of 0.8 to 1 (*Hassan & Subasi, 2017*), indicates strong agreement between the model's classification results and those of certified sleep experts, further validating the model's accuracy.

Furthermore, the model's performance was evaluated on the ISRUC-Sleep dataset, and the results are presented in Table 3. The model achieves high F1-scores of 88.98%, 82.50%, 89.25%, and 83.46% for W, N2, N3, and REM stages, respectively, while the F1-score for the N1 stage is relatively low at 55.53%. The overall Acc and MF1 are 81.97% and 79.94%, respectively, with the $\kappa$ value of 0.766. It should be noted that the model's classification performance on the ISRUC-Sleep dataset is lower than that of the Sleep-EDF Expanded dataset, which could be attributed to the presence of sleep disorder cases in the ISRUC-Sleep dataset. This finding suggests that sleep staging may be more challenging for individuals with sleep disorders.

## Confusion matrix

The classification results on the Sleep-EDF Expanded dataset are presented using a confusion matrix on the left side of Fig. 5A, where each row corresponds to the true sleep stage label, and each column represents the predicted sleep stage label. The diagonal elements signify the correctly classified instances, while the off-diagonal elements indicate misclassifications. The results reveal that the diagonal elements are much larger than other values in the same rows or columns. The normalized confusion matrix displays the same results but in percentage form, where each cell represents the proportion of instances classified into that cell, as illustrated on the right side of Fig. 5B). It provides additional information on the relative proportions of correctly classified instances and

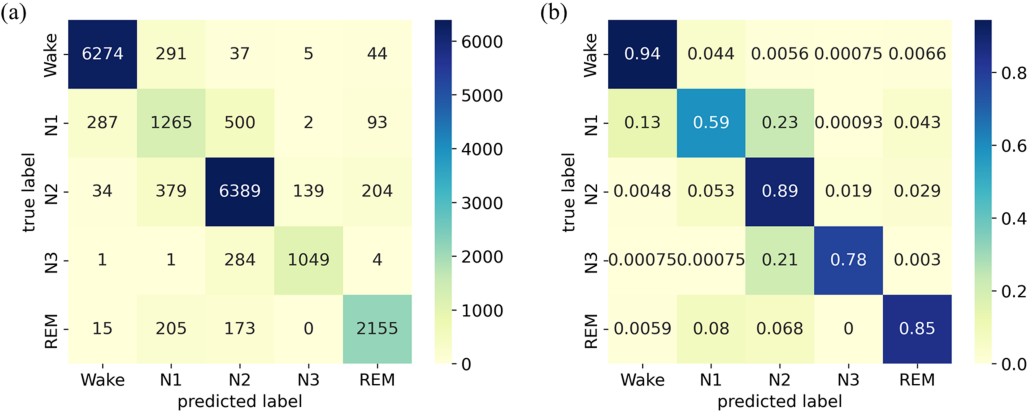

**Figure 5** (A) Confusion matrix; (B) Normalized confusion matrix on the Sleep-EDF Expanded dataset.

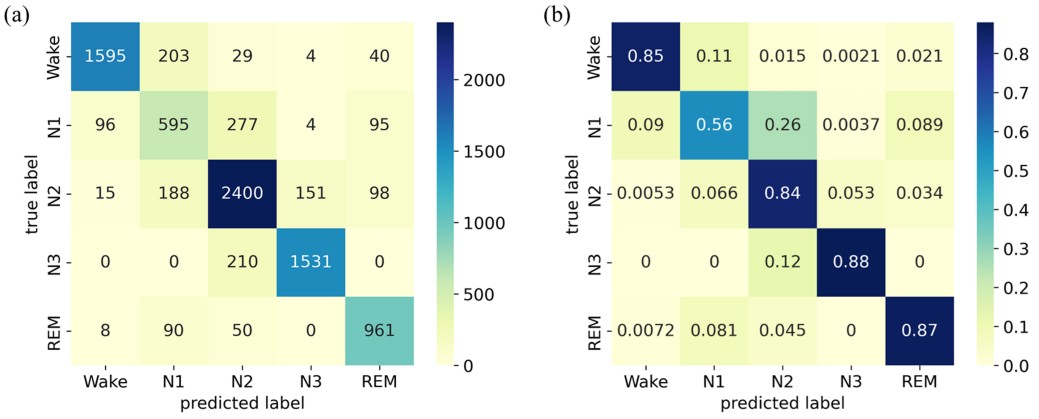

**Figure 6** (A) Confusion matrix; (B) Normalized confusion matrix on the ISRUC-Sleep dataset.

misclassifications, facilitating a more detailed assessment of the model's performance. The results indicate that the model performs well in identifying the W stage, achieving an accuracy of 94%, while its recognition of the N1 stage is limited, with an accuracy of only 59%. The model exhibits moderate performance in identifying the N2, N3, and REM stages, with an accuracy ranging from 78% to 89%. Notably, the N1 stage is frequently misclassified as the W and N2 stages, potentially due to its transitional nature between wakefulness and sleep. Additionally, several samples in the N3 stage are misclassified as the N2 stage.

The confusion matrix and its normalized version for the ISRUC-Sleep dataset are presented in Fig. 6. The results indicate that the model achieves high accuracy for the W, N2, N3, and REM stages, ranging from 84% to 88%. However, the accuracy in identifying the N1 stage is relatively low, with a score of only 56%.

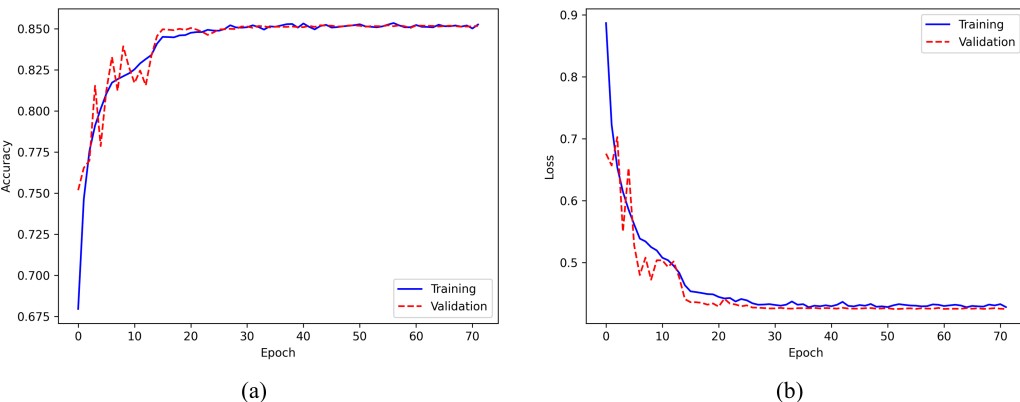

**Figure 7** (A) Accuracy curve; (B) Error curve across training epochs in training and validation set on the Sleep-EDF Expanded dataset.

## Accuracy and loss curves across epochs

The evaluation of the model's performance on the Sleep-EDF Expanded dataset during the training process is based on the accuracy and loss curves presented in Fig. 7. The accuracy curve, displayed in the left panel of Fig. 7A, illustrates the model's accuracy across the training epochs. The solid blue and dashed red curves represent the training and validation sets, respectively. The model's accuracy increases gradually with the number of epochs and stabilizes after around 20 epochs, indicating stable convergence. On the other hand, the loss curve, shown in the right panel of Fig. 7B, depicts the training and validation losses, with the solid blue and dashed red curves representing the training and validation sets, respectively. The loss value continuously decreases as the number of epochs increases and flattens after approximately 20 epochs. To prevent overfitting, the training process is terminated at the 72nd epoch using the callback of early stopping, as the validation loss stabilized for 20 epochs. Notably, the similarity between the final accuracy of the training and validation sets indicates the absence of overfitting, which is consistent with the loss curve.

Figure 8 displays the training accuracy and loss curves for the ISRUC-Sleep dataset. The accuracy increases progressively with the number of epochs and becomes stable after about 30 epochs, while the loss value decreases consistently and flattens out after around 30 epochs. To avoid overfitting, the training process is stopped at the 82nd epoch.

## Loss function parameter

The study investigated the impact of different weight coefficient combinations on the performance of the proposed model using the Sleep-EDF Expanded dataset. Four weight coefficient combinations were evaluated: [1, 1, 1, 1, 1], [1, 3, 1, 5, 3], [1, 1.5, 1, 1, 1], and [1, 2, 1, 1, 1], with the first assigning equal weight coefficients of 1 to all classes and the second assigning weight coefficients based on the number of samples in each class. The third and fourth combinations assign a weight coefficient of 1.5 and 2 to the N1 stage, respectively, while assigning a weight coefficient of 1 to the remaining sleep stages. The

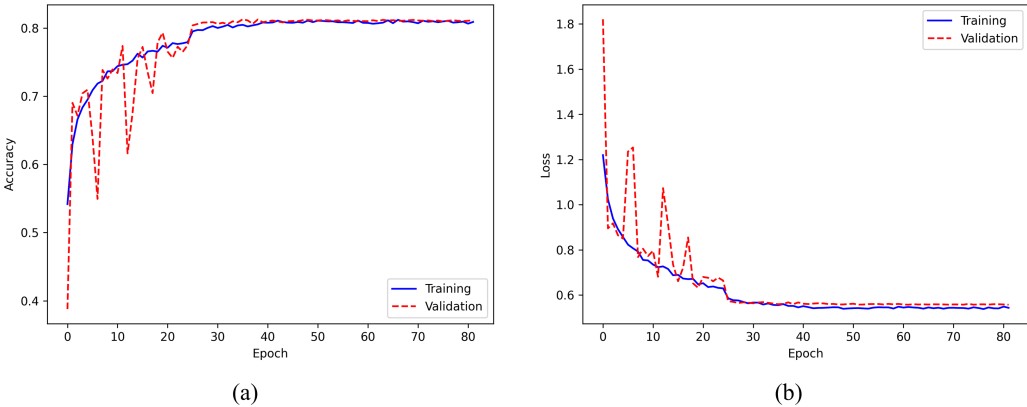

**Figure 8** (A) Accuracy curve; (B) Error curve across training epochs in training and validation set on the ISRUC-Sleep dataset.

**Table 4** Different weight coefficient combinations applied on the Sleep-EDF Expanded dataset.

| Coefficient | Overall performance | | | Per-class performance (F1-score) | | | | |
|---|---|---|---|---|---|---|---|---|
| | Acc | MF1 | $\kappa$ | W | N1 | N2 | N3 | REM |
| 1,1,1,1,1 | 86.34 | 81.13 | 0.809 | 94.56 | 55.01 | 87.82 | 83.27 | 84.97 |
| 1,3,1,5,3 | 83.23 | 79.33 | 0.775 | 93.93 | 57.93 | 83.41 | 76.94 | 84.43 |
| 1,1.5,1,1,1 | **86.39** | 81.95 | **0.812** | **94.62** | 59.00 | **87.95** | 82.79 | **85.38** |
| 1,2,1,1,1 | 85.86 | **82.15** | 0.806 | 94.53 | **60.22** | 87.45 | **83.37** | 85.18 |

**Notes.**
[1] The highest performing indices are indicated in bold font in this and subsequent tables.

results of these weight coefficient combinations applied to the Sleep-EDF Expanded dataset are presented in Table 4. Comparing the first and second combinations, it can be observed that assigning weights based solely on the number of samples per class may result in a performance deficit for both overall and per-class performance, although the performance of the N1 stage improves. In contrast, the third and fourth combinations outperform the first combination in both overall and per-class performance to the same extent. Specifically, the third combination, which assigns a weight coefficient of 1.5 to the N1 stage and 1 to the remaining four stages, balances high N1 performance and acceptable overall performance. Consequently, the study adopted this weight coefficient combination in its analysis.

Regarding the ISRUC-Sleep dataset, four different weight coefficient combinations were evaluated: : [1, 1, 1, 1, 1], [1.5, 2.5, 1, 1.5, 2.5], [1, 1.5, 1, 1, 1], and [1, 2, 1, 1, 1]. The second combination assigns weight coefficients based on the number of samples in each class. The results are presented in Table 5, where it can be observed that the highest overall metrics and per-class F1-score of N2, N3, and REM stage are achieved by assigning a weight coefficient of 1.5 to the N1 stage and a weight coefficient of 1 to the remaining four stages.

## Sequence length

We investigated the influence of varying sequence lengths of EEG epochs used as input to the model on its performance concerning the Sleep-EDF Expanded dataset. Specifically,

**Table 5** Different weight coefficient combinations applied on the ISRUC-Sleep dataset.

| Coefficient | Overall performance | | | Per-class performance (F1-score) | | | | |
|---|---|---|---|---|---|---|---|---|
| | Acc | MF1 | κ | W | N1 | N2 | N3 | REM |
| 1,1,1,1,1 | 81.86 | 78.96 | 0.764 | **89.85** | 52.00 | 81.99 | 88.89 | 82.07 |
| 1.5,2.5,1,1.5,2.5 | 79.88 | 78.20 | 0.742 | 88.16 | 54.97 | 78.95 | 88.08 | 80.86 |
| 1,1.5,1,1,1 | **81.97** | **79.94** | **0.766** | 88.98 | 55.53 | **82.50** | **89.25** | **83.46** |
| 1,2,1,1,1 | 80.57 | 78.93 | 0.749 | 88.57 | **55.67** | 80.91 | 88.86 | 80.63 |

**Notes.**
The highest performing indices are indicated in bold font.

**Table 6** Different sequence lengths applied on the Sleep-EDF Expanded dataset.

| Length | Overall performance | | | Per-class performance (F1-score) | | | | |
|---|---|---|---|---|---|---|---|---|
| | Acc | MF1 | κ | W | N1 | N2 | N3 | REM |
| 3 | 84.64 | 79.76 | 0.788 | 94.06 | 54.52 | 86.36 | 80.95 | 82.91 |
| 5 | 84.86 | 80.35 | 0.792 | 94.12 | 55.37 | 86.47 | 82.62 | 83.18 |
| 10 | 84.93 | 80.25 | 0.793 | 93.71 | 54.75 | 86.85 | 81.34 | 84.61 |
| 15 | **86.39** | **81.95** | **0.812** | **94.62** | **59.00** | **87.95** | 82.79 | **85.38** |
| 20 | 85.97 | 81.66 | 0.808 | 94.61 | 56.63 | 87.18 | **85.30** | 84.57 |

**Notes.**
The highest performing indices are indicated in bold font.

**Table 7** Different sequence lengths applied on the ISRUC-Sleep dataset.

| Length | Overall performance | | | Per-class performance (F1-score) | | | | |
|---|---|---|---|---|---|---|---|---|
| | Acc | MF1 | κ | W | N1 | N2 | N3 | REM |
| 3 | 79.51 | 78.09 | 0.736 | 90.34 | **56.19** | 78.91 | 84.74 | 80.28 |
| 5 | 79.39 | 77.15 | 0.732 | 89.65 | 51.65 | 79.62 | 85.52 | 79.34 |
| 10 | 79.78 | 77.04 | 0.739 | 90.42 | 50.73 | 78.11 | 88.69 | 77.24 |
| 15 | **81.97** | **79.94** | **0.766** | 88.98 | 55.53 | **82.50** | **89.25** | **83.46** |
| 20 | 81.81 | 79.55 | **0.766** | **91.51** | 54.88 | 79.84 | 88.10 | 83.41 |

**Notes.**
The highest performing indices are indicated in bold font.

epoch lengths of 3, 5, 10, 15, and 20 were evaluated, and the results are presented in Table 6. The findings suggest that increasing the input sequence length provides the model with more temporal information, improving overall and per-class performance. However, using longer epochs may reduce the number of available samples for model training. Given this trade-off, a sequence length of 15 was recommended for the current study.

We also examined the impact of different EEG epoch sequence lengths on the model's performance in the ISRUC-Sleep dataset. The results are presented in Table 7.

## Ablation study

The proposed model incorporates ResNet, LSTM, and SE components with the weighted cross-entropy (WCE) loss function. To evaluate the contribution of each component to the model's performance, we conducted the ablation study on the Sleep-EDF Expanded dataset, as detailed in Table 8. Specifically, we compared the classification performance of

**Table 8   Ablation study conducted on the Sleep-EDF Expanded dataset.**

| Combination | Overall performance | | | Per-class performance (F1-score) | | | | |
|---|---|---|---|---|---|---|---|---|
| | Acc | MF1 | κ | W | N1 | N2 | N3 | REM |
| ResNet+LSTM | 85.61 | 79.84 | 0.799 | 93.93 | 50.88 | 87.62 | 82.39 | 84.35 |
| ResNet+LSTM+SE | 86.34 | 81.13 | 0.809 | 94.56 | 55.01 | 87.82 | **83.27** | 84.97 |
| ResNet+LSTM+WCE | 85.70 | 80.77 | 0.801 | 94.19 | 55.51 | 87.68 | 82.13 | 84.35 |
| Our model | **86.39** | **81.95** | **0.812** | **94.62** | **59.00** | **87.95** | 82.79 | **85.38** |

**Notes.**
The highest performing indices are indicated in bold font.

**Table 9   Ablation study conducted on the ISRUC-Sleep dataset.**

| Combination | Overall performance | | | Per-class performance (F1-score) | | | | |
|---|---|---|---|---|---|---|---|---|
| | Acc | MF1 | κ | W | N1 | N2 | N3 | REM |
| ResNet+LSTM | 80.56 | 77.36 | 0.746 | 88.71 | 48.35 | 80.95 | 87.67 | 81.10 |
| ResNet+LSTM+SE | 81.86 | 78.96 | 0.764 | **89.85** | 52.00 | 81.99 | 88.89 | 82.07 |
| ResNet+LSTM+WCE | 80.95 | 78.86 | 0.753 | 88.74 | 54.71 | 81.16 | 88.19 | 81.50 |
| Our model | **81.97** | **79.94** | **0.766** | 88.98 | **55.53** | **82.50** | **89.25** | **83.46** |

**Notes.**
The highest performing indices are indicated in bold font.

four different combinations of components, namely ResNet+LSTM, ResNet+LSTM+SE, ResNet+LSTM+WCE, and ResNet+LSTM+SE+WCE (our model). By comparing the ResNet+LSTM combination with the ResNet+LSTM+SE combination, the results suggest that the SE component enhances the model's classification performance in both overall and per-class metrics, indicating the importance of feature recalibration or reweighting for the sleep staging task. Additionally, the comparison between the ResNet+LSTM combination and the ResNet+LSTM+WCE combination demonstrates that using the WCE loss function effectively mitigates the class imbalance problem, resulting in improved performance for the N1 stage while maintaining acceptable performance for other sleep stages and overall performance. Our proposed model outperforms the other three combinations in most performance metrics, except for the performance of the N3 stage. These findings suggest that integrating the ResNet, LSTM, and SE components along with the WCE loss function can effectively improve the accuracy of sleep staging.

Additionally, the ablation study was conducted on the ISRUC-Sleep dataset, as shown in Table 9. Similarly, the results suggest that the SE component effectively improves overall and per-class performance, while using the WCE loss function effectively mitigates the class imbalance problem.

## Comparative analysis with existing sleep staging models

We carried out the comparative analysis between the proposed model and several other deep learning-based sleep staging models, namely SleepEEGNet (*Mousavi, Afghah & Acharya, 2019*), DeepSleepNet (*Supratak et al., 2017*), TinySleepNet (*Supratak & Guo, 2020*), IITNet (*Seo et al., 2020*), and AttnSleep (*Eldele et al., 2021*), using the EEG channel Fpz-Cz in the Sleep-EDF Expanded dataset. Table 10 presents the results of our analysis,

**Table 10 Comparison between our model and existing sleep staging models on the Sleep-EDF Expanded dataset.**

| Model | Overall performance | | | Per-class performance (F1-score) | | | | |
|---|---|---|---|---|---|---|---|---|
| | Acc | MF1 | κ | W | N1 | N2 | N3 | REM |
| SleepEEGNet | 80.0 | 73.6 | 0.73 | 91.7 | 44.1 | 82.5 | 73.5 | 76.1 |
| DeepSleepNet | 82.0 | 76.9 | 0.76 | 84.7 | 46.6 | 85.9 | 84.8 | 82.4 |
| TinySleepNet | 83.1 | 78.1 | 0.77 | 92.8 | 51.0 | 85.3 | 81.1 | 80.3 |
| IITNet | 83.9 | 77.7 | 0.79 | 87.9 | 44.7 | **88.0** | **85.7** | 82.1 |
| AttnSleep | 82.9 | 78.1 | 0.77 | 92.6 | 47.4 | 85.5 | 83.7 | 81.5 |
| Our Model | **86.39** | **81.95** | **0.812** | **94.62** | **59.00** | 87.95 | 82.79 | **85.38** |

**Notes.**
The highest performing indices are indicated in bold font.

**Table 11 Comparison between our model and existing sleep staging models on the ISRUC-Sleep dataset.**

| Model | Overall performance | | | Per-class performance (F1-score) | | | | |
|---|---|---|---|---|---|---|---|---|
| | Acc | MF1 | κ | W | N1 | N2 | N3 | REM |
| U-Time | – | 77 | – | 88 | 55 | 79 | 87 | 83 |
| Our Model | **81.97** | **79.94** | **0.766** | **88.98** | **55.53** | **82.50** | **89.25** | **83.46** |

**Notes.**
The highest performing indices are indicated in bold font.

which indicate that the proposed model surpasses the other models in terms of all three overall performance metrics and per-class F1-score for the W, N1, and REM stages. Notably, our model exhibits improved performance in the N1 stage, which indicates the effectiveness of the proposed WCE loss function in addressing the class imbalance issue.

Furthermore, our model has a significantly lower number of parameters (0.28 m) compared to other models, such as DeepSleepNet (24.7 m), SleepEEGNet (2.6 m), and TinySleepNet (1.3 m), resulting in reduced computational resources and lower risks of overfitting the training data. This is further supported by the accuracy and error curves observed during training epochs, as illustrated in Figs. 7 and 8.

We conducted the comparison between our proposed model and a state-of-the-art deep learning-based sleep staging model, U-Sleep (*Perslev et al., 2019*), on the ISRUC-Sleep dataset. The results of this comparison are shown in Table 11, which demonstrate that our proposed model achieves superior performance over the U-Sleep model in terms of the MF1 and per-class F1-score for each sleep stage.

## DISCUSSION

Sleep staging is a critical part of sleep medicine as it involves classifying sleep into distinct stages using EEG signals, which enables the evaluation of sleep quality and the identification of sleep disorders. The application of deep learning techniques to automate sleep staging has gained significant attention in recent times owing to its potential to enhance the accuracy and efficiency of the process.

This study introduced a novel sleep staging model called 1D-ResNet-SE-LSTM, which predicts sleep stages in a many-to-many fashion from a sequence of EEG epochs. The model utilizes ResNet blocks as an epoch-wise feature extractor to extract relevant features from the raw EEG signals within each EEG epoch, while the LSTM network captures temporal dependencies and transition rules among EEG epochs. Moreover, the SE module was incorporated to enhance the feature representation and improve classification performance. The model was designed to be lightweight to conserve computational resources and is suitable for real-world applications. The WCE loss function was employed to address the performance imbalance, with the weight coefficients for each class determined by the number of samples and class complexity. The effectiveness and robustness of the model were verified using two datasets, namely the Sleep-EDF Expanded dataset and ISRUC-Sleep, to ensure generalizability.

The proposed model exhibits high performance for the W, N2, N3, and REM stages, with relatively low performance observed for the N1 stage. These results are consistent with the human interrater agreement, which has been found to lower performance in detecting the N1 stage compared to other sleep stages (*Malhotra et al., 2013*). The N1 stage is often misclassified as the W and N2 stages, possibly due to its transitional nature between wakefulness and sleep. In addition, some samples in the N3 stage may be inaccurately classified as the N2 stage. Nonetheless, the model demonstrates relatively high overall Acc and MF1, indicating its promise in sleep staging. Furthermore, the high $\kappa$ value indicates almost perfect agreement with certified sleep experts, further validating the model's accuracy. Notably, the similarity between the accuracy of the training and validation sets suggests that overfitting is absent, which is consistent with the loss curve. It is worth mentioning that sleep staging may be more challenging for individuals with sleep disorders.

To evaluate the effectiveness of the WCE loss function, we conducted experiments to compare the model's performance with different weight coefficient combinations. The results indicate that assigning weights solely based on the number of samples may result in an overall and per-class performance deficit, despite improving the N1 stage's performance. Thus, simply adjusting weight coefficients based on the number of samples in each class may not be sufficient to obtain optimal performance. Assigning the highest weight coefficient of 1.5 to the N1 stage while setting the weight coefficients for the other four stages to 1 achieves high N1 performance with acceptable overall performance. However, it is essential to note that some samples within a sleep stage may have varying difficulty levels for recognition. For instance, transitional epochs during sleep pose a more significant challenge for accurate classification as they exhibit features of both preceding and succeeding sleep stages, making them more ambiguous to categorize correctly (*Phyo et al., 2022*). Therefore, assigning different weights to individual samples may be beneficial better to reflect their significance in the sleep staging task.

Furthermore, we investigated the influence of varying sequence lengths of EEG epochs as input to the model's performance. Our findings reveal that increasing the input sequence length provides the model with more temporal information, resulting in improved overall and per-class performance. However, using longer epochs decreases the number of available

samples for training the model. As a result, we selected a sequence length of 15 for this study, given the trade-off between temporal information and available training data.

The ablation study was performed to evaluate the contribution of each component to the model's performance. The results suggest that the SE component improves the model's classification performance across overall and per-class metrics, underscoring the importance of feature recalibration or reweighting. Moreover, adopting the WCE loss function effectively mitigates the class imbalance problem, improving N1 stage performance while maintaining satisfactory performance for other sleep stages and overall accuracy.

Compared to other cutting-edge models, the proposed model yields superior overall performance metrics and per-class F1-scores for some sleep stages. Furthermore, the proposed model possesses considerably fewer parameters than other models, which lowers the computational resources required and minimizes the risk of overfitting the training data.

However, the proposed model is subject to several limitations. Firstly, the model is a sequence-to-sequence network, which requires input sequences of EEG epochs. Secondly, the weight coefficients in the loss function could be further optimized to balance the classification performance of each sleep stage, such as adjusting the weight coefficient for the N3 stage, or gradient re-weighting based on the difficulty level of each EEG epoch sample to classify (*Liao et al., 2021*). Finally, apart from the channel attention mechanism, incorporating spatial attention mechanisms could enhance the model's effectiveness (*Woo, Park & Lee, 2018*).

Future studies could enhance the proposed model by incorporating additional modalities, like EOG and EMG signals. Since the REM stage is characterized by the movement of the eyes and loss in muscle tone of the body core, EOG and EMG signals may provide critical information to separate the REM stage from other sleep stages (*Loh et al., 2020*). Additionally, further studies are necessary to validate the model's generalizability on larger datasets and diverse populations. Moreover, exploring alternative loss functions and architectures may further improve the accuracy of sleep staging. Furthermore, the model's interpretability needs to be explored to comprehend how it arrives at its classification decisions and identify potential errors or biases (*Phan et al., 2022*; *Jany et al., 2022*; *Horie et al., 2022*). Meanwhile, addressing the relatively low performance for the N1 stage could be an area of focus for future research.

## CONCLUSIONS

In summary, this study introduced a novel approach for sleep staging that utilizes deep learning techniques such as ResNet, LSTM, and SE components, along with the WCE loss function. The results demonstrate the effectiveness and robustness of the proposed model in classifying sleep stages. The high degree of agreement between the model's predictions and those of human sleep experts highlights its potential as a valuable tool for improving the diagnosis and treatment of sleep disorders. The proposed model can help reduce human clinicians' workload, making sleep assessment and diagnosis more effective. Furthermore, this research contributes to the emerging deep learning-based sleep staging field and provides essential insights for the sleep medicine community.

### Funding

The authors received no funding for this work.

### Competing Interests

Weiming Li and Junhui Gao are employed by Shanghai Nuanhe Brain Technology Co. Ltd. The authors declare that there are no competing interests.

### Author Contributions

- Weiming Li conceived and designed the experiments, performed the experiments, analyzed the data, performed the computation work, prepared figures and/or tables, authored or reviewed drafts of the article, and approved the final draft.
- Junhui Gao conceived and designed the experiments, performed the experiments, authored or reviewed drafts of the article, and approved the final draft.

### Data Availability

The Sleep-EDF dataset is available at Sleep-EDF Database Expanded: Available at https://physionet.org/content/sleep-edfx/1.0.0/.

The ISRUC-Sleep dataset is available at ISRUC-SLEEP Dataset: Available at https://sleeptight.isr.uc.pt/?page_id=48.

The source codes are available at GitHub and Zenodo:

- Available at https://github.com/weiming1122/1D-ResNet-SE-LSTM.

- weiming1122. (2023). weiming1122/1D-ResNet-SE-LSTM: v1.0.0 (v1.0.0). Zenodo. Available at https://doi.org/10.5281/zenodo.8065867.

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
