# Peer review of "Automatic sleep staging by a hybrid model based on deep 1D-ResNet-SE and LSTM with single-channel raw EEG signals"

_PeerJ Computer Science, doi:10.7717/peerj-cs.1561_

## Round 0.1 · original submission · Major Revisions

Please improve the article accordingly all comments of reviewers.

Reviewer 1 ·

Basic reporting

This study aimed to propose an DL approach for Sleep stage classification using EEG data. I have the following major suggestions.
What is the novelty of this study although several DL approaches for Sleep stage classification have been proposed earlier?
Please write down the contribution of the study at the end part of the Introduction section in bulleted form.
Authors should include conceptual figures of their proposed DL approach with more details and model parametrization.

Experimental design

EEG is highly sensitive to the powerline, muscular, and cardiac artifacts. In EEG data preprocessing, authors need to mention how you handle AC power, ECG, and EMG artifacts in EEG signals. Do the authors think that their proposed method is robust to such kinds of artifacts?

Validity of the findings

Both CV-training and testing ROC curves need to be shown for each sleep class. What model validation method authors used?

Additional comments

How did the authors deal with dataset class imbalance challenges in classification?
Authors need to mention the model parameters or hyperparameters of the proposed DL model.
Authors should introduce the ML/DL applications in disease, and mental workload prediction in broad scope, such as article, big-ecg: cardiographic predictive cyber-physical system for stroke management; in article, explainable artificial intelligence model for stroke prediction using eeg signal; in article, healthsos: real-time health monitoring system for stroke prognostics; in article, quantitative evaluation of task-induced neurological outcome after stroke; in article, driving-induced neurological biomarkers in an advanced driver-assistance system; and in article, sleepexplain: explainable non-rapid eye movement and rapid eye movement sleep stage classification from eeg signal.
The discussion section needs to be improved. Authors must make discussion on the advantages and drawbacks of their proposed method with other studies adding a table in the discussion section.
I recommend using deepSHAP/GradCAM to explain the contribution of EEG features in the DL models.
From the writing point of view, the manuscript must be checked for typos and the grammatical issues should be improved.

·

Basic reporting

This paper provides a concise and informative overview of the research study on sleep staging using deep learning with EEG signals. It effectively communicates the purpose, methodology, and key findings of the study. The paper is well-written and presents the methodology clearly. The authors have provided adequate results and discussion, which can serve as a valuable foundation for future research and development in this field. However, there are some suggestions to enhance the quality of this manuscript:

1. Introduction: Consider incorporating additional related research to provide a more comprehensive background and contextualize the study further.
2. Literature review: It would be beneficial to include a literature review section or additional references to establish the existing knowledge and research gaps in the field.
3. Proofreading: Please carefully proofread the manuscript for typographical and grammatical errors to ensure clarity and professionalism.

By addressing these points, the quality and comprehensiveness of the manuscript can be further improved.

Experimental design

No comment.

Validity of the findings

No comment.

Additional comments

Abstract:
Strengths:
1. The abstract provides a clear overview of the research problem and the proposed solution. It highlights the importance of sleep staging, the challenges faced in using deep learning methods with EEG signals, and the novel approach proposed in this study.
2. The abstract describes the main components of the proposed model, namely the 1D-ResNet-SE-LSTM architecture, which combines a residual convolutional neural network and a long short-term memory network. It also mentions the use of a weighted cross-entropy loss function to address class imbalance.
3. The abstract provides performance results on two publicly available datasets, Sleep-EDF Expanded and ISRUC-Sleep, including accuracy rates and F1-scores for each sleep stage. It also mentions the kappa coefficient as a measure of agreement with certified sleep experts.
4. The abstract mentions the investigation of different weight coefficient combinations and sequence lengths of EEG epochs, as well as an ablation study to evaluate the contribution of each component to the model's performance. This demonstrates a comprehensive analysis of the proposed model.

Shortcomings:
1. The abstract could provide more specific information about the datasets used, such as the number of subjects, the recording duration, and any demographic characteristics. This would help readers assess the generalizability of the results.
2. While the abstract mentions that the proposed model outperforms existing sleep staging models, it does not provide details on the specific models or their performance metrics. Including this information would add more context and highlight the significance of the improvement.
3. The abstract does not mention any limitations or potential shortcomings of the proposed model or the study itself. It would be helpful to include a brief discussion of the limitations to provide a balanced perspective.

Overall, the abstract effectively presents the research problem, the proposed model, and the evaluation results. It could benefit from providing more specific information about the datasets and including a discussion of limitations.

---

## Round 0.2 · accepted · Accept

The authors made all corrections and well improved the article.

·

Basic reporting

No comment.

Experimental design

No comment.

Validity of the findings

No comment.